# Pediatric anti-N-methyl-D-aspartate receptor (NMDAR) encephalitis: Exploring psychosis, related risk factors, and hospital outcomes in a nationwide inpatient sample: A cross-sectional study

**Sanobar Jaka**[1]*, **Sukhnoor Singh**[2], **Sreshatha Vashist**[3], **Sandesh Pokhrel**[4], **Ericka Saldana**[5], **Albulena Sejdiu**[6], **Sanjana Taneja**[7], **Abimbola Arisoyin**[8], **Raja Mogallapu**[9], **Sasidhar Gunturu**[10,11], **Anil Bachu**[12], **Rikinkumar S. Patel**[13]

1 Department of Population Health, Section on Tobacco, Alcohol and Drug Use, New York University Grossman School of Medicine, New York, NY, United States of America, 2 Department of Psychiatry, Sri Guru Ram Das Institute of Medical Sciences and Research, Amritsar, Punjab, India, 3 Department of Psychiatry, N.C. Medical College and Hospital, Panipat, Haryana, India, 4 Department of Psychiatry, Nepal Medical College, MBBS, Attarkhel, Kathmandu, Nepal, 5 Department of Psychiatry, Salvadoran University Alberto Masferrer, San Salvador CP, El Salvador, 6 Department of Psychiatry, St. Chyril and Methodius St Chyril and Methodius, Skopje, North Macedonia, 7 Department of Psychiatry, Lady Hardinge Medical College, New Delhi, India, 8 Department of Psychiatry College of Medicine, Psychiatry Department Idiaraba, University of Lagos, Lagos, Nigeria, 9 Department of Psychiatry, West Virginia University School of Medicine, Martinsburg, WV, United States of America, 10 Department of Psychiatry, Bronxcare Health System, Bronx, NY, United States of America, 11 Department of Psychiatry, Icahn School of Medicine at Mount Sinai, New York, NY, United States of America, 12 Department of Psychiatry, Baptist Health System—University of Arkansas for Medical Sciences, Little Rock, AR, United States of America, 13 Department of Psychiatry & Behavioral Sciences, Duke University School of Medicine, Durham, NC, United States of America

* sj2900@nyu.edu

## Abstract

### Objective

Our study aims to examine the risk factors for comorbid psychosis in pediatric patients hospitalized for anti-N-methyl-d-aspartate receptor (NMDAR) encephalitis and its impact on hospital outcomes.

### Methods

We conducted a cross-sectional study using the nationwide inpatient sample (NIS 2018–2019). We included 3,405 pediatric inpatients (age 6–17 years) with a primary discharge diagnosis of anti-NMDAR encephalitis. We used binomial logistic regression model to evaluate the odds ratio (OR) of variables (demographic and comorbidities) associated with comorbid psychosis.

### Results

The prevalence of comorbid psychosis in anti-NMDAR encephalitis inpatients was 5.3%, and majorly constituted of adolescents (72.2%) and females (58.3%). In terms of race,

**Data Availability Statement:** All relevant data are housed within the manuscript. Data set used to run the analysis is available upon request from the corresponding author as well as publicly available from NIS at https://hcup-us.ahrq.gov/nisoverview.jsp

**Funding:** the author(s) received no specific funding for this work.

**Competing interests:** The authors have declared that no competing interests exist.

Blacks (OR 2.41), and Hispanics (OR 1.80) had a higher risk of comorbid psychosis compared to Whites. Among comorbidities, encephalitis inpatients with depressive disorders (OR 4.60), sleep-wake disorders (OR 3.16), anxiety disorders (OR 2.11), neurodevelopmental disorders (OR 1.95), and disruptive behavior disorders (OR 2.15) had a higher risk of comorbid psychosis. Anti-NMDAR encephalitis inpatients with comorbid psychosis had a longer median length of stay at 24.6 days (vs. 9.8 days) and higher median charges at $262,796 (vs. $135,323) compared to those without psychotic presentation.

## Conclusion

Adolescents, females, and Blacks with encephalitis have a higher risk of psychotic presentation leading to hospitalization for anti-NMDAR encephalitis. Identification of demographic predictors and comorbidities can aid in early recognition and intervention to optimize care and potentially reduce the healthcare burden.

## Introduction

Encephalitis is a condition characterized by inflammation of the brain parenchyma, leading to changes in mental state and psychiatric symptoms. If left untreated in the early stages, it can result in long-term morbidity in up to 50% of patients [1]. With the advancement of diagnostic techniques, the diagnosis of encephalitis has become more accurate in recent years [1]. This led to a striking increase in the worldwide incidence of encephalitis by 12.5% with the number of reported cases increasing from 1,284,160 in 1990 to 1,444,720 in 2019 [2]. And, the reported incidence of encephalitis among hospitalized patients is estimated to be approximately 3–7 cases per 100,000 [3, 4]. While encephalitis can affect individuals of all ages, it is more commonly observed in children [5]. From 2000 to 2010, the incidence rate of encephalitis in the US was 7.3 cases per 100,000 person-years, peaking in infants at 13.5 cases per 100,000 person-years, and reaching its lowest point in youth aged 10–14 years at 4.1 cases per 100,000 person-years [6].

Encephalitis has diverse etiologies involving two main forms: primary infectious encephalitis, caused by direct pathogen invasion of the central nervous system (CNS), and immune-mediated encephalitis, resulting from immune system damage. Viruses can invade the CNS through various means, infecting neurons and causing cytotoxicity. Pathogens can also induce inflammation or vasculitis, leading to tissue damage or ischemia [6]. Encephalitis has a variety of causes, including infectious, autoimmune, and unknown factors. Viral infection stands as the predominant cause in most cases followed by autoimmune causes [7].

Autoimmune encephalitis is associated with a range of disorders and has been observed in patients diagnosed with schizophrenia, bipolar disorder and depression [8]. There have been reports that previous research has indicated that patients initially diagnosed with bipolar disorder, autistic traits, or psychotic disorder may develop autoimmune encephalitis several years later highlighting the possibility of misdiagnosis and mistreatment of autoimmune encephalitis with predominant psychiatric manifestations [9–12]

Anti-N-methyl-d-aspartate receptor (NMDAR) encephalitis is commonly characterized by a relapsing pattern, with patients often experiencing persistent behavioral, memory, cognitive, and executive function alterations even during the recovery phase [13]. In contrast, α-amino-3-hydroxy-5-methyl-4-isoxazolepropionic acid receptor (anti-AMPA-receptor) encephalitis typically leads to a rapid deterioration of neurological function [14].

A longitudinal study conducted at a single center in China found that the typical onset age for encephalitis is 21 years, with a noteworthy 31.4% of affected patients being under 18 years old [15]. This emphasizes the critical importance of early detection and management of encephalitis in young individuals. The study also shed light on the clinical manifestations of anti-NMDAR encephalitis, with psychosis being the most prevalent symptom observed in 182 patients (82.7%), closely followed by seizures reported in 178 patients (80.9%) [15]. These findings underscore the significance of recognizing and addressing the psychological and neurological symptoms of anti-NMDAR encephalitis.

The available research on encephalitis and comorbid psychosis in pediatric inpatients is scarce. In order to overcome this limitation, we conducted a cross-sectional study based on inpatient data to assess the risk of psychiatric comorbidities among pediatric inpatients diagnosed with anti-NMDAR encephalitis and study the prevalence of psychosis in anti-NMDAR encephalitis with its risk factors and impact on hospital outcomes using nationwide inpatient sample.

## Method and materials

### Study sample

We conducted a cross-sectional study using the nationwide inpatient sample (NIS 2018–2019) which is derived from non-federal community hospitals from 48 and the District of Columbia in the US [16].

Our study sample included 3,405 pediatric inpatients (age 6–17 years) with a primary discharge diagnosis of anti-NMDAR encephalitis using the international classification of diseases, tenth revision (ICD-10) diagnostic code G04.0x to G04.9x. The study sample was further grouped based on co-diagnosis of psychotic disorders using ICD-10 codes (F22 or F29).

### Variables

The demographic variables included in the study were age, sex, race/ethnicity, and median household income. The race/ethnicities included in the study are White, Black, Hispanic, Asian/Pacific Islander, Native American and others. Comorbidities are co-diagnoses in the patient records and we included anxiety disorders, obsessive compulsive-related disorders [17] (OCD), trauma-and stressor-related disorders, neurodevelopmental disorders, disruptive behavior disorders (DBD, including disruptive, impulse-control and conduct disorders), and alcohol, opioid, cannabis, sedative and stimulant-related disorders, suicidal behaviors (ideation/attempt), epilepsy and cerebral palsy (CP).

The length of stay (LOS) during hospitalization for primary diagnosis i.e. encephalitis in this study was all-cause. The charges during hospitalization do not include professional fees and non-covered charges.

### Statistical analysis

We compared the distributions of sociodemographic characteristics, and comorbidities in anti-NMDAR encephalitis inpatients with psychotic disorders and other psychiatric illnesses using descriptive statistics and Pearson's chi-square test. Next, we used the logistic regression model to evaluate the odds ratio (OR) of variables associated with psychotic disorders to evaluate the sociodemographic predictors and comorbid risk factors of anti-NMDAR encephalitis. Lastly, we analyzed the comorbidities that increase the risk of hospitalization for psychotic disorders by age groups (6–11, 12–17 and 18–24) in demographic-adjusted independent binomial logistic regression models. All analyses were conducted using a statistical package for the social

sciences (SPSS, IBM Corp., Armonk, NY) and statistical significance was set at a two-sided P value <0.05.

## Results

The total number of inpatients with anti-NMDAR encephalitis included in the study was 3405, out of which 180 (5.3%) had comorbid psychotic disorders. The majority of the study inpatients were young adults, 6–11 years (52%), male (54.2%) and White (50.7%).

The mean age of the patients with comorbid psychotic disorders was significantly higher at 12.8 years compared to 11 years for those without. Additionally, a significantly higher percentage of patients with comorbid psychotic disorders were in the age range of 12–17 years (72.2%) compared to those without (46.7%). Regarding sex, a significantly lower percentage of male patients had comorbid psychotic disorders (41.7%) compared to females (58.3%). There was a significant difference in race between the two groups, with a higher percentage of black patients in the comorbid psychotic disorders group (21.2%) compared to those without (13.2%)

There were no significant differences in median household income between the two groups. Anti-NMDAR encephalitis inpatients with psychotic disorders had significantly higher prevalence of comorbid depressive disorders, anxiety disorders, OCD, neurodevelopmental disorders, DBD, and sleep-wake disorders compared to those without (P<0.001 for all).

Regarding hospital outcomes, anti-NMDAR encephalitis inpatients with comorbid psychotic disorders had a significantly longer median LOS at 10 days compared to 5 days for those without. Patients with comorbid psychotic disorders also had significantly higher median charges at $137,862 compared to $61,003 for those without (P<0.001) as shown in Table 1.

The risk of having comorbid psychotic disorders in anti-NMDAR encephalitis was higher in adolescents (12–17 years) compared to those aged 6–11 years (OR 2.41, 95% CI 1.65–3.51). The odds were also higher in female patients compared to males (OR 1.53, 95% CI 1.07–2.20). In terms of race, Blacks (OR 2.41, 95% CI 1.49–3.87, p<0.001), and Hispanics (OR 1.80, 95% CI 1.16–2.79, p = 0.009) with anti-NMDAR encephalitis had higher odds of comorbid psychotic disorders compared to the Whites.

Among comorbidities, anti-NMDAR encephalitis inpatients with depressive disorders (OR 4.60, 95% CI 2.89–7.34), sleep-wake disorders (OR 3.16, 95% CI 2.12–4.73), anxiety disorders (OR 2.11, 95% CI 1.40–3.19), neurodevelopmental disorders (OR 1.95, 95% CI 1.30–2.92), and DBD (OR 2.15, 95% CI 1.19–3.91) had a higher risk of comorbid psychotic disorders. However, inpatients with comorbid epilepsy did not have an increased association (OR 0.62, 95% CI 0.42–0.92) for comorbid psychotic disorders as shown in Table 2.

## Discussion

About five percent of the pediatric inpatients hospitalized for anti-NMDAR encephalitis had comorbid psychotic disorders. This was prevalent in adolescents, and females. Blacks and Hispanics had about two-fold higher risk of having a psychotic presentation with anti-NMDAR encephalitis. Furthermore, pediatric inpatients with anti-NMDAR encephalitis and psychiatric comorbidities including depressive and anxiety disorders, OCD, neurodevelopmental disorders, DBD, and sleep-wake disorders had a greater risk of psychotic presentation during hospitalization stay.

Autoimmune encephalitis presentations associated with psychotic symptoms differ in terms of age and sex. When limbic encephalitis is prevalent in elderly patients, then anti-NMDA-R encephalitis with psychotic symptoms is predominant in females, and children

**Table 1. Variations in demographic characteristics, comorbid conditions, and clinical outcomes among hospitalized individuals with encephalitis with comorbid psychotic disorders.**

| Variable | Total | Comorbid psychotic disorders | | P value |
|---|---|---|---|---|
| | | No | Yes | |
| Number of inpatients | 3405 | 3225 | 180 | - |
| Mean age (SD) | 11.1 (3.4) | 11.0 (3.4) | 12.8 (3.1) | 0.016 |
| Age, in % | | | | |
| 6–11 years | 52.0 | 53.3 | 27.8 | <0.001 |
| 12–17 years | 48.0 | 46.7 | 72.2 | |
| Sex, in % | | | | |
| Male | 54.2 | 54.9 | 41.7 | 0.001 |
| Female | 45.8 | 45.1 | 58.3 | |
| Race, in % | | | | |
| White | 50.7 | 51.0 | 45.5 | 0.020 |
| Black | 13.6 | 13.2 | 21.2 | |
| Hispanic | 23.3 | 23.3 | 24.2 | |
| Other | 12.4 | 12.5 | 9.1 | |
| Median household income, in % | | | | |
| Below 50th percentile | 45.6 | 45.5 | 47.2 | 0.659 |
| Above 50th percentile | 54.4 | 54.5 | 52.8 | |
| Comorbidities, in % | | | | |
| Depressive disorders | 5.0 | 3.7 | 27.8 | <0.001 |
| Anxiety disorders | 16.0 | 14.6 | 41.7 | <0.001 |
| Obsessive compulsive -related disorders | 5.1 | 4.8 | 11.1 | <0.001 |
| Neurodevelopmental disorders | 22.6 | 21.7 | 38.9 | <0.001 |
| Disruptive behavior disorders | 3.7 | 3.3 | 11.1 | <0.001 |
| Sleep-wake disorders | 9.5 | 8.5 | 27.8 | <0.001 |
| Epilepsy | 31.6 | 31.8 | 27.8 | 0.261 |
| Hospital outcomes | | | | |
| Median length of stay, in days (IQR) | - | 5 (9) | 10 (23) | <0.001 |
| Median charges, in $ (IQR) | - | 61003 (104939) | 137862 (240917) | <0.001 |

IQR: interquartile range

[18, 19]. In our study, comorbid psychotic disorder was more common in adolescents and females and the risk increased by 2.4 and 1.5 times respectively compared to their counterparts.

Cohen and Marino conducted a comprehensive nationwide epidemiological study that demonstrated significantly higher lifetime rates of psychotic symptom-related disorders (15.3%) among Blacks. Latino Americans had a slightly lower prevalence of 13.6%, followed by White Americans at 9.7% and Asian Americans at 9.6% [20]. Although there is consistent evidence of a higher burden of psychoses among Blacks, there is ongoing research to understand the racial disparities in diagnosis and access to care by race/ethnicities [21]. In our study we found that Blacks (by 2.4 times) and Hispanics (by 1.8 times) had a higher risk of comorbid psychotic disorders compared to Whites. However, we did not find strong evidence or correlations with specific ethnicities in the literature to compare our findings, possibly due to the various etiologies of encephalitis.

A notable finding in our study was the association of the spectrum of psychiatric comorbidities and anti-NMDAR encephalitis with a psychotic presentation in pediatric inpatients. Anti-

**Table 2. Predictive factors indicating comorbid psychotic disorders among hospitalized individuals with encephalitis.**

| Variable | Odds ratio | 95% Confidence Interval | P value |
|---|---|---|---|
| Age | | | |
| 6–11 years | Reference | | |
| 12–17 years | 2.41 | 1.65–3.51 | <0.001 |
| Sex | | | |
| Male | Reference | | |
| Female | 1.53 | 1.07–2.20 | 0.021 |
| Race | | | |
| White | Reference | | |
| Black | 2.41 | 1.49–3.87 | <0.001 |
| Hispanic | 1.80 | 1.16–2.79 | 0.009 |
| Others | 1.26 | 0.69–2.32 | 0.456 |
| Median household income | | | |
| Below 50th percentile | Reference | | |
| Above 50th percentile | 0.85 | 0.59–1.19 | 0.345 |
| Comorbidities | | | |
| Depressive disorders | 4.60 | 2.89–7.34 | <0.001 |
| Anxiety disorders | 2.11 | 1.40–3.19 | <0.001 |
| Obsessive compulsive-related disorders | 1.34 | 0.72–2.51 | 0.361 |
| Neurodevelopmental disorders | 1.95 | 1.30–2.92 | 0.001 |
| Disruptive behavior disorders | 2.15 | 1.19–3.91 | 0.012 |
| Sleep-wake disorders | 3.16 | 2.12–4.73 | <0.001 |
| Epilepsy | 0.62 | 0.42–0.92 | 0.017 |

NMDAR encephalitis inpatients with comorbid depressive and anxiety disorders, and OCD had about two to four times higher risk of psychotic disorders. Comorbid neurodevelopmental disorders (by two times), DBD (by two times) and sleep-wake disorders (by three times) increased the risk of psychotic presentation in encephalitis. The identification of autoimmune encephalitis subtypes has renewed interest in the involvement of glutamatergic receptors and associated proteins in psychiatric disorders [22]. Several antigen targets are proteins or receptors that play a crucial role in memory, cognition, behavior, and the development of psychotic symptoms [23]. Dysregulation of the glutamatergic system, particularly NMDAR hypofunction, has been proposed as a central mechanism in schizophrenia [24]. Antagonism of the NMDA receptor in healthy individuals can induce cognitive and motor disturbances similar to those observed in psychotic disorders [25]. Experimental data also suggest that elevated dopamine levels in psychotic disorders could be a result of NMDAR hypofunction [25]. The existence of rare cases of purely psychiatric forms of NMDAR-Ab encephalitis suggests that some individuals with psychotic symptoms may have an underlying encephalitic condition and could potentially benefit from immunomodulatory treatments [25]. Our findings call for more immunologic studies to understand the conundrum of these comorbidities with psychosis and encephalitis.

This study represents a novel investigation into the prevalence of psychosis in hospitalized encephalitis patients in the United States, filling a significant knowledge gap. Previous literature has primarily focused on case series exploring autoimmune encephalitis and its presentation as psychosis [26–28]. However, it is important to acknowledge certain limitations of our study, such as the inability to categorize specific types of encephalitis and the observational nature of the study design, which prevents us from establishing causal associations. When the

signs and symptoms of psychosis are part of primary diagnosis of encephalitis, often clinician may fail to add a co-diagnosis of unspecified psychotic disorder and that may result in under-representation of comorbid psychotic disorders in our study sample. The NIS dataset offers several strengths that contribute to its value as a research resource [16]. Firstly, its large sample size provides a substantial number of hospital admissions, enabling researchers to obtain statistically significant results and make reliable inferences. Secondly, the dataset covers a wide geographic area, ensuring national representation and enhancing the generalizability of findings. Additionally, we have used ICD-10 diagnostic codes to identify anti-NMDAR encephalitis, psychotic disorders and other psychiatric comorbidities, and these codes are billing/administrative codes applied by clinicians at the discharge of patient. Moreover, the NIS dataset encompasses diverse patient populations, facilitating investigations into healthcare utilization and outcomes among different demographic groups [16]. However, it is essential to recognize the limitations of the NIS dataset. Firstly, the absence of individual patient identifiers restricts the ability to track patients across multiple admissions or link data with other sources, limiting longitudinal analysis at the individual level. Secondly, the accuracy of the data is dependent on the coding practices of healthcare providers, which may introduce coding errors or inconsistencies. Additionally, the NIS dataset primarily consists of administrative data, lacking detailed clinical information that could provide deeper insights into patient conditions and treatment outcomes [16].

## Conclusions

Adolescents, females, and Blacks with anti-NMDAR encephalitis have a higher risk of psychotic presentation leading to hospitalization. These risk factors should be considered by healthcare providers when assessing pediatric patients with anti-NMDAR encephalitis, emphasizing the need to consider the possibility of psychotic symptoms, particularly within these vulnerable groups. Moreover, the study highlights the importance of recognizing preexisting psychiatric comorbidities, such as depressive and anxiety disorders, OCD, neurodevelopmental disorders, disruptive behavior disorders, and sleep-wake disorders, as they significantly increase the risk of presenting with psychotic symptoms in anti-NMDAR encephalitis cases. Consequently, it is crucial to screen for these psychiatric conditions when managing pediatric anti-NMDAR encephalitis patients to identify individuals who are at a higher risk of developing psychotic symptoms. Identification of demographic predictors and comorbidities can aid in early recognition and intervention to optimize care and potentially reduce the healthcare burden.

## Author Contributions

**Conceptualization:** Rikinkumar S. Patel.

**Data curation:** Sanobar Jaka, Sukhnoor Singh, Sreshatha Vashist, Sandesh Pokhrel, Ericka Saldana, Albulena Sejdiu, Sanjana Taneja, Abimbola Arisoyin, Rikinkumar S. Patel.

**Formal analysis:** Rikinkumar S. Patel.

**Project administration:** Sanobar Jaka.

**Supervision:** Raja Mogallapu, Sasidhar Gunturu, Anil Bachu.

**Writing – original draft:** Sanobar Jaka, Sukhnoor Singh, Sreshatha Vashist, Sandesh Pokhrel, Ericka Saldana, Albulena Sejdiu, Sanjana Taneja, Abimbola Arisoyin, Raja Mogallapu, Sasidhar Gunturu, Anil Bachu, Rikinkumar S. Patel.

**Writing – review & editing:** Sanobar Jaka, Sreshatha Vashist, Sandesh Pokhrel, Ericka Saldana, Albulena Sejdiu, Sanjana Taneja, Abimbola Arisoyin, Raja Mogallapu, Sasidhar Gunturu, Anil Bachu, Rikinkumar S. Patel.

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
