## [Decision Letter · Decision Letter 0]

10 Sep 2023

PONE-D-23-23242Predictive Risk factors and Impact of Psychosis in Pediatric Encephalitis: A Cross-national Hospital StudyPLOS ONE

Dear Dr. Jaka,

Thank you for submitting your manuscript to PLOS ONE. After careful consideration, we feel that it has merit but does not fully meet PLOS ONE’s publication criteria as it currently stands. Therefore, we invite you to submit a revised version of the manuscript that addresses the points raised during the review process.

We look forward to receiving your revised manuscript.

Kind regards,

Meisam Akhlaghdoust, M.D., M.P.H.

Academic Editor

PLOS ONE

Journal Requirements:

Reviewers' comments:

Reviewer's Responses to Questions

**Comments to the Author**

1. Is the manuscript technically sound, and do the data support the conclusions?

Reviewer #1: Yes

Reviewer #2: Partly

2. Has the statistical analysis been performed appropriately and rigorously? 

Reviewer #1: Yes

Reviewer #2: No

3. Have the authors made all data underlying the findings in their manuscript fully available?

Reviewer #1: Yes

Reviewer #2: Yes

4. Is the manuscript presented in an intelligible fashion and written in standard English?

Reviewer #1: Yes

Reviewer #2: Yes

5. Review Comments to the Author

Reviewer #1: Thanks the authors to work on this valuable project. However, there are some comments as follows:

Title: It is not a prediction model study. Just you want to evaluate the related factors. Thus, it is better to revise the title to “Psychosis in Pediatric Encephalitis and its related factors and impact on hospital outcomes: A Cross-sectional study using nationwide inpatient sample (NIS 2018-2019)”

In the whole manuscript, please avoid using predictive risk factors. Use risk factors or related factors.

Abstract:

-Revise the aim according to the recommended title.

We conducted a cross-sectional study was conducted using the nationwide inpatient sample (NIS 2018-2019).Omit “was conducted”.

-We used logistic regression model to evaluate the odds ratio (OR) of variables (demographic and comorbidities) associated with comorbid psychosis. It is better to use multivariable logistic regression model to evaluate the OR and 95% CI.

-“. In terms of race, Blacks (OR 2.41), and Hispanics (OR 1.80) had a higher likelihood of comorbid psychosis compared to Whites. Among comorbidities, encephalitis inpatients with depressive disorders (OR 4.60), sleep-wake disorders (OR 3.16), anxiety disorders (OR 2.11), neurodevelopmental disorders (OR 1.95), and disruptive behavior disorders (OR 2.15) had a higher likelihood of comorbid psychosis. Encephalitis inpatients with comorbid psychosis had a longer mean length of stay at 24.6 days (vs. 9.8 days) and higher mean charges at $262,796 (vs. $135,323) compared to those without psychotic presentation”. Due to rarity assumption, it is better use risk instead of likelihood. Please consider this issue in the whole manuscript. In addition, for length of stay and charges, due to the skewed nature of the variable, please report median (IQR) instead of mean (SD).

-Conclusion : Adolescents, females, and Blacks with encephalitis have a higher likelihood of psychotic presentation leading to hospitalization. Identification of demographic predictors and comorbidities can aid in early recognition and intervention to optimize care and potentially reduce the healthcare burden. Please use “ have a higher risk” not likelihood.

Introduction:

-Revise the aim according to the recommended title

Methods:

-Write this section according to the STROBE writing standard.

-Stat. analysis: please use multivariable logistic regression

-Ethical approval: it is a major concern that you did not have any approval REC. I think it is secondary data analysis not a primary study.

Results:

-Line 142- 161: please avoid mentioning the results both in table and text. It is recommended to revise this section.

-Table : please use median (IQR) for length of stay and charges due to the skewed nature of these variable.

-Line 167-174: Please avoid using likelihood. It is recommended to use risk instead due to the rarity assumption of the outcome. Also, please do not repeat the results of table 2 in text.

-“However, patients 183 with comorbid epilepsy had lower (OR 0.62, 95% CI 0.42-0.92, p=0.017) compared to those 184 without epilepsy in encephalitis inpatients as shown in Table 2” had lower what?

-It is recommended to review the results completely and rewrite it.

Conclusion:

-“Adolescents, females, and Blacks with encephalitis have a higher likelihood of 249 psychotic presentation leading to hospitalization”. Please use “Higher risk” instead of likelihood.

Reviewer #2: I am honored to have the opportunity to review “Title: Predictive Risk factors and Impact of Psychosis in Pediatric Encephalitis: A Cross-national Hospital Study”. This work includes a significant number of patients. However, I have a few comments and questions:

Abstract:

-In aim of the study, the authors aimed to evaluate the impact of hospital outcomes, while in the title they mentioned “Impact of Psychosis on Pediatric Encephalitis”.

-In line 53, it seems that a space is needed between patients. Encephalitis in patients with comorbid psychosis had a longer mean length of stay at 24.6 days.

-In conclusion, how identification of demographic predictors and comorbidities can aid in early recognition and intervention to optimize care and potentially reduce the healthcare burden?

Methods:

-Patients with encephalitis have been investigated for a year. It is not mentioned by which criteria psychosis was diagnosed. It is not clear which signs and symptoms are considered as psychosis. As the patients with encephalitis may have related symptoms such as hallucinations, delirium and other symptoms during the course of the disease.

-Hospital outcomes included hospital stay and charges. These are not impacts of psychosis on patients. On the other hand, due to the existence of more comorbidities in groups, how can they estimate the impact of psychosis alone?

Results:

-The data in text and tables are replicated. It is better to summarize the text and merge the tables.

-The etiology of encephalitis is not explained in this study, which is a major risk factor for encephalitis consequences.

Discussion:

-In line 192, Autoimmune encephalitis presentations associated with psychotic symptoms differ in terms of age and sex. When limbic encephalitis is prevalent in elderly (male or female? ) patients, then anti-NMDA-R encephalitis with psychotic symptoms is predominant in females, and children.

6. PLOS authors have the option to publish the peer review history of their article (what does this mean?). If published, this will include your full peer review and any attached files.

Reviewer #1: **Yes: **Mahin Nomali, MSCCN, Ph. D in Epidemiology

Reviewer #2: No

---

## [Author Response · Author response to Decision Letter 0]

22 Nov 2023

Meisam Akhlaghdoust, M.D., M.P.H.

Academic Editor

PLOS ONE

We express our gratitude to all the reviewers for their thorough and comprehensive peer review. We have implemented the required revisions in the manuscript, and as a result, we believe our work has been significantly improved. Kindly review our responses and the corresponding alterations in the manuscript below.

Reviewer #1: Thanks, the authors to work on this valuable project. However, there are some comments as follows:

- Title: It is not a prediction model study. Just you want to evaluate the related factors. Thus, it is better to revise the title to “Psychosis in Pediatric Encephalitis and its related factors and impact on hospital outcomes: A Cross-sectional study using nationwide inpatient sample (NIS 2018-2019)” 

Response: The title has been revised in the view of other changes in the manuscript.

- In the whole manuscript, please avoid using predictive risk factors. Use risk factors or related factors. 

Response: This has been addressed.

- Abstract:

Revise the aim according to the recommended title. 

Response: We have fixed the title to appropriately reflect the aims and the meaning of the whole study.

- We conducted a cross-sectional study was conducted using the nationwide inpatient sample (NIS 2018-2019).Omit “was conducted”. 

Response: thank you for pointing out, this has been fixed.

- We used logistic regression model to evaluate the odds ratio (OR) of variables (demographic and comorbidities) associated with comorbid psychosis. It is better to use multivariable logistic regression model to evaluate the OR and 95% CI. – 

Response: we have used binomial logistic regression model based on our goals and the nature of the data, and it has been added in line 131

- In terms of race, Blacks (OR 2.41), and Hispanics (OR 1.80) had a higher likelihood of comorbid psychosis compared to Whites. Among comorbidities, encephalitis inpatients with depressive disorders (OR 4.60), sleep-wake disorders (OR 3.16), anxiety disorders (OR 2.11), neurodevelopmental disorders (OR 1.95), and disruptive behavior disorders (OR 2.15) had a higher likelihood of comorbid psychosis. Encephalitis inpatients with comorbid psychosis had a longer mean length of stay at 24.6 days (vs. 9.8 days) and higher mean charges at $262,796 (vs. $135,323) compared to those without psychotic presentation”. Due to rarity assumption, it is better use risk instead of likelihood. Please consider this issue in the whole manuscript. In addition, for length of stay and charges, due to the skewed nature of the variable, please report median (IQR) instead of mean (SD). 

Response: Likelihood has been replaced with risk as suggested. We have also reported median instead of mean. Line 57 – 60 

- Conclusion : Adolescents, females, and Blacks with encephalitis have a higher likelihood of psychotic presentation leading to hospitalization. Identification of demographic predictors and comorbidities can aid in early recognition and intervention to optimize care and potentially reduce the healthcare burden. 

Response: Please use “ have a higher risk” not likelihood. – this has been fixed. Line: 61 – 62

Introduction:

- Revise the aim according to the recommended title

Response: We have revised the title and aim to reflect the aim of the study.

- Methods: Write this section according to the STROBE writing standard. 

Response: we have followed the required methodological subsection based on the external data NIS HCUP that has been used in this study, and also complies with journal guidelines.

- Stat. analysis: please use multivariable logistic regression.

Response: we have used binomial logistic regression model based on our goals and the nature of data, and it has been added in line 131

- Ethical approval: it is a major concern that you did not have any approval REC. I think it is secondary data analysis not a primary study. – 

Response: as mentioned in the manuscript The NIS is a publicly available de-identified dataset from the agency for healthcare research and quality (AHRQ). So, as per the US Department of Health and Human Services, institutional review board permission was not required. Ethical review and approval was not required for the study on human participants in accordance with the local legislation and institutional requirements. Written informed consent from the participants' legal guardian/next of kin was not required to participate in this study in accordance with the national legislation and the institutional requirements. More information on this can be found on - https://hcup-us.ahrq.gov/db/nation/nis/NIS_Introduction_2019.jsp.

Results:

- Line 142- 161: please avoid mentioning the results both in table and text. It is recommended to revise this section. – 

Response: We strongly feel this imperative for audience who may not have complete knowledge to interpret the results suing the tables alone. We have described the important results only in the text form.

- Table : please use median (IQR) for length of stay and charges due to the skewed nature of these variable. 

Response: this has been added

-Line 167-174: Please avoid using likelihood. It is recommended to use risk instead due to the rarity assumption of the outcome. Also, please do not repeat the results of table 2 in text. – 

Response: likelihood has been changed in the whole manuscript. As we mentioned earlier, we feel it’s imperative to mention results in the text as well as tables.

-“However, patients 183 with comorbid epilepsy had lower (OR 0.62, 95% CI 0.42-0.92, p=0.017) compared to those 184 without epilepsy in encephalitis inpatients as shown in Table 2” had lower what?

-It is recommended to review the results completely and rewrite it. – 

Response: we have revised the results section as recommended.

Conclusion:

-“Adolescents, females, and Blacks with encephalitis have a higher likelihood of 249 psychotic presentation leading to hospitalization”. Please use “Higher risk” instead of likelihood. – 

Response: this has been fixed.

Reviewer #2: I am honored to have the opportunity to review “Title: Predictive Risk factors and Impact of Psychosis in Pediatric Encephalitis: A Cross-national Hospital Study”. This work includes a significant number of patients. However, I have a few comments and questions:

Abstract:

-In aim of the study, the authors aimed to evaluate the impact of hospital outcomes, while in the title they mentioned “Impact of Psychosis on Pediatric Encephalitis”. 

Response - we have revised the aim as well as the title.

-In line 53, it seems that a space is needed between patients. Encephalitis in patients with comorbid psychosis had a longer mean length of stay at 24.6 days. – 

Response: we meant to say “inpatients”

-In conclusion, how identification of demographic predictors and comorbidities can aid in early recognition and intervention to optimize care and potentially reduce the healthcare burden? 

Response: In essence, this process works by utilizing demographic predictors and comorbidity assessment to identify individuals at risk or in need of specific healthcare interventions at an earlier stage. By doing so, healthcare providers can offer timely and tailored care, addressing health issues before they escalate and potentially become more costly to treat. This proactive approach helps improve patient outcomes and, in the long run, may alleviate the overall strain on the healthcare system by reducing the burden of managing advanced or preventable health conditions.

Methods:

-Patients with encephalitis have been investigated for a year. It is not mentioned by which criteria psychosis was diagnosed. It is not clear which signs and symptoms are considered as psychosis. As the patients with encephalitis may have related symptoms such as hallucinations, delirium and other symptoms during the course of the disease. 

Response: addressed this in methods and limitation section of discussion – line #109 - #111 and #231- 251 in limitations section. We also specified the type of encephalitis and revised the manuscript and title accordingly.

-Hospital outcomes included hospital stay and charges. These are not impacts of psychosis on patients. 

Response: agree with this. The results clearly define the differences in LOS and charges in NMDAR encephalitis w and w/o psychosis.

On the other hand, due to the existence of more comorbidities in groups, how can they estimate the impact of psychosis alone?

Response: logistic regression model as mentioned in table 2 is utilized to estimate these risk factors including demographics.

-The data in text and tables are replicated. It is better to summarize the text and merge the tables.

Response: Table 1 and 2 are different analytical models and they cannot be merged. It is the style of original paper writing where some significant parts of tables have been described in the result and we have made necessary edits

-The etiology of encephalitis is not explained in this study, which is a major risk factor for encephalitis consequences.

Response: addressed this in methods and clarified on anti-NMDAR encephalitis type

Discussion:

-In line 192, Autoimmune encephalitis presentations associated with psychotic symptoms differ in terms of age and sex. When limbic encephalitis is prevalent in elderly (male or female? ) patients, then anti-NMDA-R encephalitis with psychotic symptoms is predominant in females, and children. – 

Response: Reference paper did not mention any gender in elderly in limbic encephalitis. We tried to look to look up other information and papers but did not find that information.

SUPPORTING INFO is added at end of paper. 

Response: As a researcher, we would need to comply with the rules and regulations related to the use of the HCUP NIS database. These agreements typically include restrictions on the sharing of the raw data with anyone who has not obtained the necessary permissions and licenses to access the data. Instead, we have already provided a detailed description of my methods and analysis, including any relevant codes or algorithms, in order to allow other researchers to replicate or validate my results. It is important to balance the need for transparency and replication in scientific research with the need to protect the privacy and confidentiality of individuals whose data is included in the HCUP NIS database. Anyone can access the data and replicate my findings given they have received necessary data usage agreements from HCUP NIS.

---

## [Editor Report · Decision Letter 1]

20 Dec 2023

Pediatric anti-N-methyl-D-aspartate receptor (NMDAR) encephalitis: Exploring psychosis, related risk factors, and hospital outcomes in a nationwide inpatient sample: a cross-sectional study

PONE-D-23-23242R1

Dear Dr. Jaka,

We’re pleased to inform you that your manuscript has been judged scientifically suitable for publication and will be formally accepted for publication once it meets all outstanding technical requirements.

Kind regards,

Meisam Akhlaghdoust, M.D., M.P.H.

Academic Editor

PLOS ONE
---

## [Editor Report · Acceptance letter]

1 Feb 2024

PONE-D-23-23242R1 

PLOS ONE

Dear Dr. Jaka, 

I'm pleased to inform you that your manuscript has been deemed suitable for publication in PLOS ONE. Congratulations! Your manuscript is now being handed over to our production team.

Kind regards, 

on behalf of

Dr. Meisam Akhlaghdoust 

Academic Editor

PLOS ONE